# Can We Debiase Multimodal Large Language Models via Model Editing?

## ABSTRACT

Multimodal large language models (MLLM) have been observed to exhibit biases originating from their training datasets. Unlike unimodal LLMs, biases in MLLMs may stem from interactions between multiple modalities, which increases the complexity of multimodal debiasing. Conventional approaches like fine-tuning to alleviate biases in models are costly and data-hungry. Model editing methods, which focus on post-hoc modifications of model knowledge, have recently demonstrated significant potential across diverse applications. These methods can effectively and precisely adjust the behavior of models in specific knowledge domains, while minimizing the impact on the overall performance of the model. However, there is currently no comprehensive study to drive the application of model editing methods in debiasing MLLM and to analyze its pros and cons. To facilitate research in this field, we define the debiasing problem of MLLM as an editing problem and propose a novel set of evaluation metrics for MLLM debias editing. Through various experiments, we demonstrate that: (1) Existing model editing methods can effectively alleviate biases in MLLM and can generalize well to semantically equivalent image-text pairs. However, most methods tend to adversely affect the stability of the MLLM. (2) Compared to editing the visual modality of the MLLM, editing the textual modality yields better results in addressing MLLM biases. (3) Model editing based debiasing method can achieve generalization across different types of biases.

## CCS CONCEPTS

• **Computing methodologies** → **Artificial intelligence**.

## KEYWORDS

Multimodal Large Lanuage Model, Multimodal Debiasing, Model Editing

## 1 INTRODUCTION

Large language models have emerged as a pivotal and versatile component in a variety of user-facing language technologies due to their outstanding performance (Chowdhery et al. [9], OpenAI [37], Touvron et al. [47], *inter alia*). Multimodal Large Language Model (MLLM) takes a step forward from LLM by utilizing powerful large language models as the brain to perform multimodal tasks [52]. Specifically, MLLM typically consists of three key elements:

*ACM MM, 2024, Melbourne, Australia*
© 2024 Copyright held by the owner/author(s). Publication rights licensed to ACM.
ACM ISBN 978-x-xxxx-xxxx-x/YY/MM
https://doi.org/10.1145/nnnnnnn.nnnnnnn

an LLM-based text encoder as the brain, an image encoder to receive multimodal information, and a bridge to establish effective connections from the two encoders (like Perceiver Resampler in Flamingo [2]). The remarkable emergent capabilities exhibited by MLLM, such as zero-shot image-to-text generation and OCR-free math reasoning, are seldom observed within conventional methodologies, signifying a potential avenue towards the attainment of artificial general intelligence[26, 28, 58].

Similar to the LLM, the MLLM still inadvertently and unavoidably acquires biased information embedded within its extensive corpus, leading to negative stereotypes and social biases encoded within the model. For instance, the MLLM has shown tendencies to associate images of white individuals with higher-status categories. Besides, in ambiguous professional contexts, the MLLM manifests a predisposition to associate male images with male-dominated professions (such as doctors, construction workers, etc.) more than female-dominated professions [15]. Furthermore, the MLLM evinces biases towards specific demographic groups. For example, attributes most associated with Islam and Judaism might encompass terms linked to poverty, terrorism, and extremism, which carry extremely negative connotations [22]. As biased MLLMs are applied more extensively in the real world, they can generate extremely detrimental social impacts and result in discriminatory treatment against the population groups they impact.

Currently, numerous studies are dedicated to the pursuit of constructing fair and unbiased neural networks, aiming to ensure equitable distribution of benefits across diverse segments of society. These studies can be roughly categorized into three main paradigms: (1) Modifying the dataset distribution before training by balancing groups of samples with and without bias, e.g., via data augmentation [17] or sample synthesis [5, 12, 25]. (2) Strategies based on model outputs to address fairness issues, namely identifying and mitigating social biases without the need for further weight optimization or dataset manipulation [48]. (3) Explicitly eliminating the influence of biases during the model training or inference process [20, 27, 40]. However, when it comes to mitigating specific biases in MLLM, such as reducing biases between gender and occupation, these three paradigms fail to directly generate fair models through new training stages or optimization processes [3, 16, 50, 51]. Specifically, the first stream of work is often insufficient to produce fair neural models for MLLM, because even if the data perfectly represents population distributions, undesirable characteristics such as societal stereotypes and biases can still be present [39]. The second stream of work does not truly address the bias encoded in the MLLM, potentially leading to non-robustness [40]. The third stream of work typically requires a large amount of training, which, for MLLM, incurs prohibitively high computational costs due to the large amounts of parameters. Besides, involving the training process can alter the pre-trained weights with no constraints, which risks losing valuable existing knowledge in the MLLM [14, 21].

Therefore, methods for mitigating feature or prediction biases, independent of the availability of non-biased data, are preferable in the MLLM.

Recently, model editing [11, 32–34] involves post-training adjustments to alter the factual knowledge stored in the model, has shown potential in addressing these issues. The objective of model editing is to modify a model's behavior in specific knowledge domains effectively and targetedly, thereby enabling it to generate more accurate and relevant outputs while ensuring the stability of its overall performance. Moreover, a series of studies have begun applying model editing methods to specific downstream tasks, such as editing personality [30], natural language inference [1], etc. Besides, Yan et al. [49] formulate social debiasing as an editing problem, and employ various model editing methods on unimodal LLMs for bias mitigation. It indicates that existing model editing methods can effectively preserve knowledge and mitigate biases in unimodal LLMs.

However, unlike editing knowledge in the unimodal LLM, in the case of the MLLM, biased outputs stem from the synergistic effects of various modalities. For example, biased outputs may originate not only from LLM but also from human-like errors involving image information, such as misunderstandings or misrecognition (e.g., color blindness or color weakness can affect color recognition in images). Consequently, exploring how to employ model editing techniques to eliminate biases present in the MLLM is a worthwhile field of inquiry.

To facilitate research in this area, we conduct a comprehensive study on model editing based debiasing methods for the MLLM. Specifically, following Yan et al. [49] and Cheng et al. [7], we first expand the prior evaluation principles of model editing to multimodal debiasing settings, including **Reliability**, **Generality** and **Locality**. Then, according to these evaluation principles, we further construct a benchmark for model editing in MLLM debiasing, which includes two subtasks: Visual Question Answering (VQA) and Image Captioning (IC). In more detail, for the reliability evaluation, we first conduct rigorous data filtering, selecting data that performed poorly for MLLM to create dedicated reliable debiasing editing datasets. For the generality evaluation, we divide it into text and multimodal generality, and use OpenAI's gpt-3.5-turbo-instruct and Stable Diffusion [41] to generate rephrased text and rephrased images. For the locality evaluation, similar to generality, we partition it into text and multimodal locality to assess the stability of MLLM across both text and multimodal datasets.

By utilizing two widely-used MLLM models, BLIP-2 [26] and MiniGPT-4 [58], we conduct a comprehensive debiasing assessment on a range of model editing methods, such as SERAC [35] and in-context knowledge editing [55]. Experimental results indicate that debiasing editing methods for MLLM are effective in reducing model biases, with most methods achieving edit success rates close to 100%. However, some of these methods come at the expense of sacrificing other aspects of MLLM capabilities. Besides, we examine how debiasing editing, when applied individually to the textual and visual module of MLLM, affects model biases. The results indicate that editing the textual module within MLLM is more effective in comparison to editing the visual module. Additionally, we also analyze the performance of multimodal model editing methods

on specific biases and whether they could achieve generalization across bias types.

In summary, the primary contributions of this work are:

(1) To our best knowledge, we take the first step to explore the influence of editing on the internal biases within the MLLM.
(2) We introduce a novel benchmark for debiasing editing in MLLM, which can be used for evaluating the reliability, locality, and generality of model editing based debiasing methods via the image captioning task and the visual question answering task.
(3) We further investigate the impact of editing various modules of MLLM on biases within the model, and explore the generality of MLLM debiasing editing across different types of biases.

## 2 RELATED WORK

### 2.1 Multimodal Large Language Models

In recent years, significant progress has been made in the development of large language models, achieving remarkable emergent abilities by expanding both data and model sizes. While LLMs have shown surprising zero/few-shot inference performance across many natural language processing tasks, they inherently lack the ability to understand visual information since they can only understand text. Meanwhile, large vision foundation models have made rapid progress in visual perception. As a complement, LLMs and visual models have converged towards each other, giving rise to the new domain of Multimodal Large Language Models (MLLM). The introduction of the MLLM paradigm has alleviated the substantial computational costs incurred by the ever-expanding scale of models and datasets during traditional multimodal model training. Building upon the foundations of LLMs and visual foundation models, MLLMs can accept inputs from multiple senses, enabling more flexible interactions with users. Additionally, MLLMs more accurately reflect how humans perceive the world. Furthermore, as a more comprehensive task solver, compared to LLMs, MLLMs typically support a broader range of tasks. The debut of GPT-4 (Vision) [37] and Gemini has left a remarkable impression on the understanding and generation abilities of multimodal (MM) models, igniting a research frenzy in MM-LLM. Initially, research on MLLMs primarily focused on multimodal content understanding (e.g., visual question answering) and text generation (e.g., image-to-text comprehension), exemplified models like BLIP-2 [26], LLaVA [28], MiniGPT-4 [58]. Later, the functionality of Multimodal LLMs is expanded to support specific modality outputs (e.g., image-text output), exemplified models like GILL [24], MiniGPT-5 [56]. In this paper, we study the biases of MLLMs in image captioning and visual question answering tasks, using BLIP-2 OPT and MiniGPT-4 as our base models.

### 2.2 Multimodal Debiasing

Recent studies have found that multimodal models exhibit biases originating from their training datasets. Utilizing biased multimodal models in real-world applications may result in adverse consequences. Therefore, addressing the societal biases present in MLLM and mitigating their negative impacts in the application process

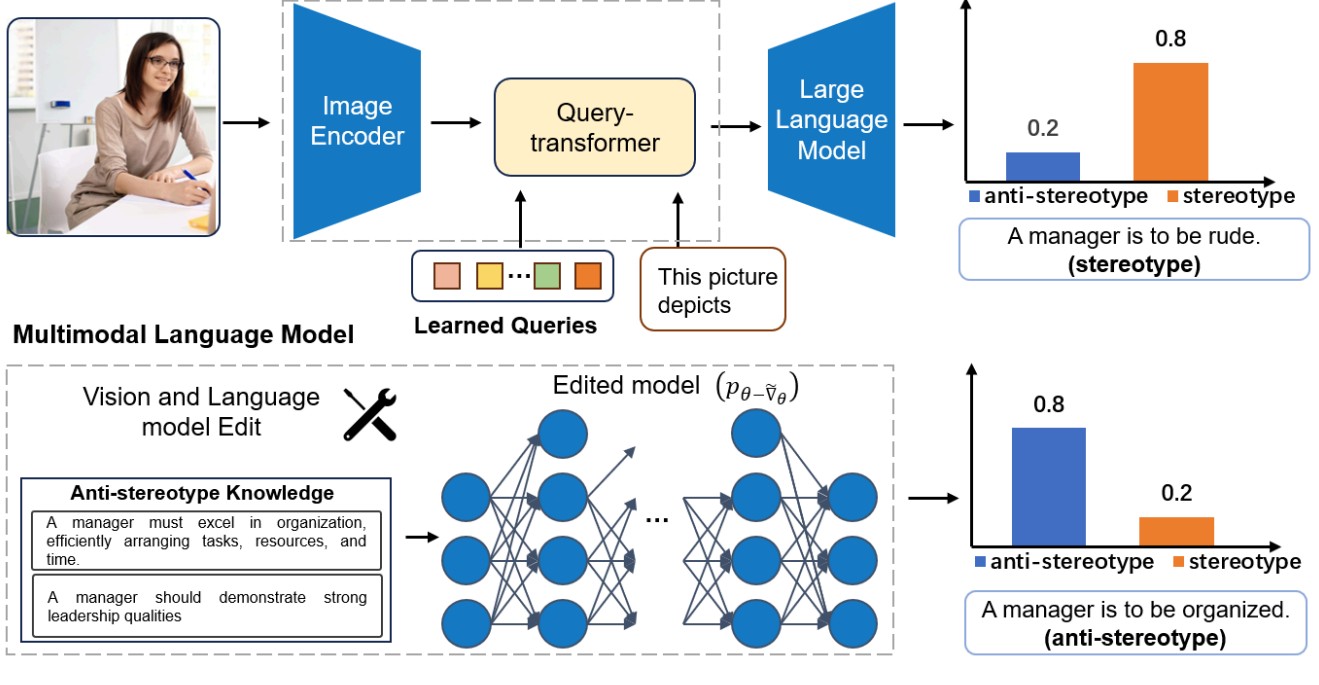

**Figure 1: Illustration of MLLM (e.g., BLIP-2 OPT) debiasing editing. MLLM can be divided into visual and textual modules. The debiasing editing of MLLM involves applying multimodal model editing methods to these modules, increasing the probability of unbiased knowledge in image-text pairs to mitigate biases present in the model.**

are essential prerequisites for future exploration and deployment of MMLM. Strategies to mitigate bias in MLLMs can be classified based on various stages of the model workflow: (1) Preprocessing techniques are designed to detect and eliminate biases and unfairness in the dataset early on, by modifying the distribution of the dataset [6, 12, 36, 50]. (2) Optimization methods during training include generating fair models for specific tasks using a single optimization algorithm and conducting new training stages to rectify existing biased models [3, 20, 27, 38, 40, 46, 50]. (3) Post-processing techniques mainly entail adjusting model outputs to mitigate bias and unfairness, aiming to detect and eliminate social biases without directly accessing the model itself, thus without needing additional weight optimization or dataset manipulation [13, 42, 48]. However, how to effectively mitigate biases in pre-trained MLLMs while minimizing disruption to model capabilities and performance remains extensively unexplored.

## 2.3 Model Editing

LLMs have demonstrated extraordinary abilities in understanding and generating text. The continuous dynamic update of world knowledge necessitates ongoing updates to LLMs to correct outdated information or integrate new knowledge, ensuring their sustained relevance. Furthermore, many applications also require continual adjustments to the model to address defects or errors present within it. Therefore, how to efficiently and lightly modify LLMs in real time has garnered increasing attention. Recently, model editing techniques for LLMs have seen significant development [18, 44, 53].

Model editing methods are designed to quickly and precisely modify LLMs, allowing them to generate more accurate and pertinent outputs. These methods can be broadly classified into two main types: intrinsic methods, which involve modifying the model architecture or parameters to edit the intrinsic knowledge of the model, and extrinsic methods, which resort to external knowledge to adjust the model input or output space. Intrinsic methods require the model to modify its own parameters to master new knowledge. The conventional method of updating knowledge involves fine-tuning the model, which requires a considerable amount of computational resources. Besides, fine-tuning often leads to catastrophic forgetting and overfitting. Apart from fine-tuning methods, some approaches have tried to use knowledge-specific methods to modify the model weights, which can be divided into two categories: meta-learning [11, 19, 45] and location-then edit [29, 32, 33]. The former doesn't directly update the model's weights but teaches a hyper-network to learn the changes $\Delta W$ of the model. The latter explores how models store knowledge, based on some mechanisms derived from LLMs, to locate the storage position of knowledge and then edit specific areas. It also adopts causal analysis methods to detect which part of the hidden state is more important. This direct editing of model parameters provides a more persistent solution for changing the model's behavior. However, further research is needed due to the unclear mechanisms of LLMs. Extrinsic methods learn representations of new knowledge and merge this information with the representations of the original model. Very recently, Cheng et al. [7] extended model editing from single-modal to multimodal and

demonstrated some effectiveness. Based on this, we select several model editing methods suitable for multimodal tasks to explore the impact of multimodal debiasing editing on multimodal model bias.

## 3 DEBIASING MLLM VIA MODEL EDITING

Our goal is to conduct a comprehensive analysis of multimodal model debiasing editing. In this section, we first introduce the task definition of MLLM edit in section §3.1. Subsequently, we propose three metrics for evaluating MLLM debiasing editing in section §3.2. Based on these three metrics, we detail the construction process of our dataset in section §3.3. Finally, we introduce the MLLMs (§3.4) and baseline methods (§3.5) utilized in our experiments.

### 3.1 Preliminary: MLLM Editing

Model editing methods are primarily used for knowledge editing. The purpose of model editing is to modify a model into a new one, covering some of the original knowledge to achieve the desired output while preserving the integrity of the model's other knowledge. Let $\theta$ denote a MLLM, with $\theta_{vision}$ and $\theta_{text}$ representing its visual and textual components, respectively. Specifically, given an image input $img$ and a text prompt input $text$, an editing method $f$ edits the multimodal MLLM's output from original output $y_o$ to the target output $y_e$.

$$y_o = \arg\max_y(p(y|img, text; \theta))$$
$$\theta_e = f(\theta, y_o, y_e) \tag{1}$$

where we refer $\theta_e$ as the model after edit. Therefore, for MLLM, a successful model editing should modify the model's knowledge to produce the desired output $y_e$.

$$y_e = \arg\max_y(p(y|img, text; \theta_e)) \tag{2}$$

Following Cheng et al. [7], the requirements for model editing should also meet the criterion of Generality and Locality. The generalization capability of the MLLM is reflected in the ability of the modified MLLM $\theta_e$ to yield the target output $y_e$ for any rephrased images and text. The locality metrics for model editing in MLLM aim to minimize any unforeseen side effects on the broader knowledge base of MLLM caused by model editing, ensuring the stability of the model. Specifically, MLLM should also satisfy that for any broader knowledge $text$ and $img$, the modified model's output remains consistent with the output of the original model, as described by the following:

$$\arg\max_y(p(y|img, text; \theta_e)) = \arg\max_y(p(y|img, text; \theta))$$
$$\forall(img, text) \tag{3}$$

### 3.2 Task Definition on MLLM Debiasing Editing

In this section, we formulate the MLLM debias editing task, focusing on pairs of biased and unbiased sentences associated with images. Considering an image-text pair $(img, text, y_{more}, y_{less})$, where $y_{more}$ is a more stereotypical biased sentence compared to $y_{less}$. We argue that an MLLM exhibits bias towards this image-text pair if the likelihood of MLLM tends to prefer the biased sentence.

$$p(y_{more}|img, text; \theta_e) > p(y_{less}|img, text; \theta_e) \tag{4}$$

To attain a fairer MLLM, we can choose to decrease the likelihood of $y_{more}$ or increase the likelihood of $y_{less}$. For model editing, increasing the likelihood of $y_{less}$ is evidently a more feasible approach. Building upon the aforementioned premise, we propose the following three metrics for the comprehensive evaluation of MLLM debiasing editing.

**Reliability.** Reliability measure serves to evaluate the bias level of the model following modification. Specifically, it assesses whether the modified MLLM $\theta_e$ satisfies the following condition.

$$p(y_{more}|img, text; \theta_e) < p(y_{less}|img, text; \theta_e) \tag{5}$$

**Generality.** Merely debiasing individual image-text pairs is insufficient for the model debias editing process. We expect a fair MLLM should not only achieve debiasing effects on the original image-text pairs but also on their equivalent inputs (e.g., rephrased sentences or rephrased images), implying a degree of generalization ability in the model's debiasing process. To address this issue, we introduce two generalization sub-metrics. The first one is **T-Generality**.

$$p(y_{more}|img, text_r; \theta_e) < p(y_{less}|img, text_r; \theta_e) \tag{6}$$

where $text_r$ presents the rephrased textual prompt in IC task and the rephrased question in VQA task. It evaluates whether the likelihood of unbiased sentences generated by the edited MLLM, under the conditions of unchanged images and rephrased text, surpasses the likelihood of biased sentences. Besides, the second sub-metrics we proposed is **V-Generality**.

$$p(y_{more}|img_r, text; \theta_e) < p(y_{less}|img_r, text; \theta_e) \tag{7}$$

where $img_r$ presents the rephrased image. It evaluates if the likelihood of $y_{less}$ in the edited MLLM, with rephrased images and original text prompt, exceeds the likelihood of $y_{more}$.

**Locality.** In order to uphold model stability, it is essential to minimize the extent to which model editing affects the overall knowledge capabilities of the model. We utilize the concept of Locality to quantify this capability of the MLLM. Since most of the knowledge in MLLM is inherited from LLM, maintaining the stability of LLM is crucial. Thus, we design a **T-Locality** metric to evaluate the impact of model editing on LLM stability, as below:

$$p(y|text; \theta_e) = p(y|text; \theta) \tag{8}$$

Given that Visual block can transform images into vector representations and collaborate with natural language text during the encoding process, efforts should also be made to minimize the influence of model editing on Visual block. We define the **M-locality** as:

$$p(y|img, text; \theta_e) = p(y|img, text; \theta) \tag{9}$$

### 3.3 Debiasing Benchmark Construction

The dataset that we construct includes two sub-tasks: Image-Caption (IC) and Visual Question Answering (VQA). The former task aims to enable the MLLM to comprehend the visual content of images and generate answers based on textual questions about the image. The latter task focuses on arming the MLLM with the ability to comprehend the visual content of images and subsequently generate natural language captions for them. As mentioned in section §3.2, the content of each example in both sub-tasks datasets is denoted as $(img, text, y_{more}, y_{less})$.

**Table 1: The number of different bias types in the Reliability dataset for the Image Caption task.**

| Bias type | Age | Gender | Race | Profession | Religion |
|---|---|---|---|---|---|
| **Number** | 680 | 2262 | 3026 | 631 | 44 |

*3.3.1 Reliability Dataset Construction.* As shown in Equation 4, to benchmark our experiments, we need to construct image-text pairs containing the biased and unbiased sentences ($img, text, y_{more}, y_{less}$). The foundational data for our IC task is derived from VLStereoSet [57] and PATA dataset[43]. The former is a dataset containing biases related to Gender, Profession, Race, and Religion. It comprises a total of 1,028 image-text pairs, with images that are categorized as either stereotypical or anti-stereotypical. The latter is a dataset containing biases related to gender and racial labels as well as two age group labels (young and old). It consists of 24 scenes, each containing between 100 and 400 images, for a total of 4,934 images. As this dataset provides a set of generic biased and unbiased textual captions for each scenario, with a significantly larger number of images than captions, to ensure dataset diversity and prevent caption redundancy, for each image, we randomly select a bias type and then randomly select a sentence from the corresponding biased or unbiased caption set. In summary, our image captioning task encompasses five kinds of bias: gender, race, profession, religion, and age. The size of the proposed dataset is 6643, and the quantities of each bias type are as shown in Table 1.

The foundational data for our VQA task originates from the PAIRS dataset [15]. The PAIRS dataset comprises a collection of artificially generated human images, which are highly similar in terms of background and visual content, yet differ in aspects of gender and race. Given the ambiguity in background and visual context within the PAIRS dataset, interpretations of a subject's occupation, social standing, or intention can differ. Therefore, to obtain a biased dataset tailored to a specific model, we utilize MLLM to compute the probabilities of different labels in image-text pairs, selecting the answers having lower likelihoods as our targets for debias editing.

*3.3.2 Generality Dataset Construction.* Following the generality metric mentioned in section §3.2, we have introduced two forms of generality evaluation datasets for the MLLM. The process of constructing a general dataset is illustrated in Figure 2.

**Textual Generality Dataset.** Benefiting from the exceptional performance and remarkable problem-solving capabilities of the LLM, we can instruct the LLM to generate rephrased textual inputs by specifying task instructions. Therefore, for the VQA task, we utilize gpt-3.5-turbo-instruct to rewrite questions from the dataset. In the context of the IC task, we adopt a manually created template with 20 prompts to replace the original random prompts, inspired by Cheng et al. [7]. The concrete prompts of the IC task can be found in the Appendix.

**Visual Generality Dataset.** Diffusion models, based on the forward diffusion stage and the reverse diffusion stage, are a class of deep generative models that have achieved significant success in the field of image generation in recent years [10]. Stable Diffusion

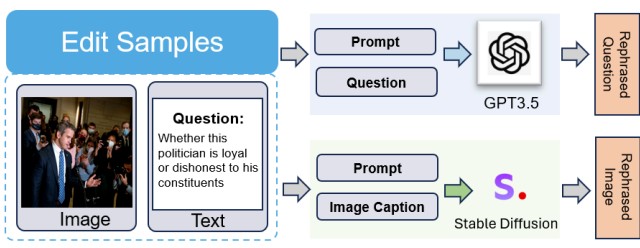

**Figure 2: Construction Process of the Generality Dataset.**

[41] is a latent text-to-image diffusion model capable of generating natural, high-quality images given textual input. We use Stable Diffusion to generate reinterpreted images. Specifically, for the IC task, we use captions in section §3.3.1 to generate reinterpreted images. For the VQA task, we leverage the prompts utilized in the PAIRS dataset for image generation to create our reinterpreted images.

*3.3.3 Locality Dataset Construction.* In order to ensure the stability of the MLLM, efforts should be made to mitigate the impact of model edits on the performance of the MLLM across broader knowledge domains. Similar to section §3.3.2, we construct two forms of datasets to evaluate the Locality of the MLLM.

**Textual Locality Dataset.** As the core of MLLM knowledge, LLM occupies a significant proportion of the parameters in MLLM. In order to gauge the stability of the LLM, we employ the Natural Questions (NQ) dataset [23] utilized in MEND. For the evaluation of locality, we calculate the KL divergence using the outputs of the MLLM before and after model editing, to facilitate constraints on model editing. To further quantify the stability of the model, the proportion of instances maintaining a top-1 status is calculated.

**Multimodal Locality Dataset.** MLLM functions through the mutual collaboration of LLM and the text module. Therefore, validating the impact of model edits on the overall performance of the Multimodal LLM is also crucial. We utilize a simple dataset, OK-VQA [31], to serve as a measure of the locality for the MLLM. Our evaluation approach to multimodal data is similar to textual locality.

## 3.4 Multimodal Large Language Models

**BLIP-2.** BLIP-2 [26] utilizes a frozen training image encoder and a frozen large-scale language model for visual-language pretraining. It employs a lightweight querying Transformer between the image encoder and the LLM, which utilizes a set of learnable query vectors to extract visual features from the frozen image encoder. BLIP-2 bridges the gap between the two modalities by training the querying Transformer only, serving as a general and efficient pretraining strategy. It can achieve state-of-the-art performance in a range of vision-language tasks. We opt for BLIP-2 OPT as our base model, which comprises a visual module consisting of ViT-L and an LLM module composed of the OPT model [54].

**MiniGPT-4.** MiniGPT-4 [58] is a powerful visual language model similar to BLIP-2, leveraging frozen visual encoders and frozen vicuna [8]. MiniGPT-4 adds a projection layer to align the encoded visual features with the Vicuna model (language model). The visual features are extracted by the pre-trained VIT-G/14 in MiniGPT-4.

**Table 2: Main results of multimodal model debias editing. Reliability denotes the probability that biases are correctly modified after editing. T-generality and V-generality represent the generality of multimodal models in text and multimodal domains. T-locality and M-locality refer to the stability of multimodal modes in text and multimodal domains.**

| Method | Image Captioning | | | | | Visual Question Answering | | | | |
|---|---|---|---|---|---|---|---|---|---|---|
| | Reliability | T-Generality | V-generality | T-Locality | M-Locality | Reliability | T-Generality | V-generality | T-Locality | M-Locality |
| | | | | | BLIP-2 OPT | | | | | |
| Base Model | 0.00 | 0.00 | 0.00 | 100.0 | 100.0 | 0.00 | 0.00 | 0.00 | 100.0 | 100.0 |
| FT-L | 71.69 | 72.61 | 70.08 | 57.31 | 10.25 | 54.19 | 51.61 | 53.55 | 62.83 | 11.68 |
| FT-V | 81.35 | 80.67 | 69.05 | **100.0** | 7.11 | 63.87 | 64.52 | 61.29 | **100.0** | 5.37 |
| IKE | 99.77 | 98.73 | 99.54 | 12.11 | 2.96 | **100.0** | **100.0** | **100.0** | 15.76 | 2.74 |
| SERAC | 98.85 | 98.73 | 98.50 | 99.98 | 10.32 | **100.0** | **100.0** | **100.0** | **100.0** | 2.58 |
| KE | 76.30 | 74.23 | 78.13 | 95.24 | 69.16 | 89.03 | 84.52 | 89.67 | 98.68 | 76.07 |
| MEND | **100.0** | **100.0** | **100.0** | 95.94 | **73.52** | 96.77 | 96.77 | 96.77 | 99.05 | **89.97** |
| | | | | | MiniGPT-4 | | | | | |
| Base Model | 84.01 | 82.54 | 76.92 | 100.0 | 15.00 | 83.23 | 53.55 | 73.55 | 100.0 | 12.74 |
| FT-L | 76.56 | 76.44 | 76.07 | 74.06 | 19.10 | 72.26 | 62.58 | 72.90 | 72.03 | 16.98 |
| FT-V | 84.01 | 82.54 | 76.92 | **100.0** | 15.00 | 83.23 | 53.55 | 73.55 | **100.0** | 12.74 |
| IKE | 97.07 | 98.17 | 95.97 | 15.61 | 4.45 | 98.71 | **100.0** | 99.36 | 15.65 | 4.26 |
| SERAC | 98.34 | 98.13 | 98.41 | 98.69 | 13.21 | **100.0** | **100.0** | **100.0** | **100.0** | 2.13 |
| KE | 89.87 | 88.89 | 89.74 | 98.69 | 69.18 | 92.26 | 90.32 | 90.32 | 98.70 | 76.99 |
| MEND | **100.0** | **100.0** | **100.0** | 98.95 | **83.41** | 96.13 | 96.13 | 89.68 | 99.36 | **83.54** |

## 3.5 Baselines

We benchmark six model editing methods, detailed as follows. These methods can be categorized into two distinct phases based on how human knowledge is acquired: (a) intrinsic methods, editing intrinsic knowledge of the model. (b) extrinsic methods, merging the knowledge into the model. [53]. The former learns representations of new knowledge and merges this information with the representations of the original model, including FT-L, FT-V, KE, and MEND. The latter requires the model to learn knowledge of its own parameters and autonomously master this knowledge, including IKE and SERAC.

**Finetuning (FT-L and FT-V).** Fine-tuning is a traditional method that involves updating model parameters to enable the model to learn target-specific knowledge. However, fine-tuning all parameters of a multimodal LLM is computationally expensive. Following Cheng et al. [7], we employ two model fine-tuning strategies. One approach is to fine-tune the last layer of the language model, denoted as FT-L, while another is to fine-tune the vision block of the multimodal LLM, which is represented as FT-V. Taking BLIP-2 OPT as an example, we fine-tune the parameters of the 31st decoder layer of the OPT model and the Q-former model, respectively.

**In-Context Knowledge Editing (IKE).** In-context Learning (ICL) is a new capability that emerged in LLMs, where language models are used to perform downstream tasks without the need for parameter updates [4]. In-context knowledge Editing (IKE) [55] helps the model generate reliable factual edits by constructing three types of demonstrations (copy, update, and retain). It first constructs a demonstration set $C = \{c_1, \ldots, c_k\}$ consisting of the training dataset. Before injecting factual knowledge $f$, it guides the model to generate appropriate answers by retrieving the most relevant demonstrations from the training set based on cosine similarity. The primary goal of IKE in knowledge editing is to maximize $p(y^* \mid x, f, C; \theta)$ when prompt $x$ is within the editing scope of the target prompt.

**SERAC.** SERAC [35] consists of an editing memory, a small auxiliary classifier, and a counterfactual model. It stores edited information in explicit memory rather than directly in the model parameters. The classifier determines whether the user's input falls within the scope of explicit memory. If the classifier identifies relevant editing examples associated with the input, it combines this example with the input and forwards it to the counterfactual model for prediction.

**Knowledge Editor (KE).** KE [11] trains a hyper-network with constrained optimization. When predicting related to edits knowledge, it utilizes the trained hyper-network to predict weight updates during testing. KE can modify facts without affecting other knowledge and achieves high computational efficiency.

**Model Editor Networks with Gradient Decomposition (MEND).** MEND [34] is an approach that learns to transform the original fine-tuning gradients into more targeted parameter updates. Specifically, MEND applies the rank-1 decomposition to partition the model into two rank-1 matrices, from which it can compute $\Delta W$, significantly reducing the number of parameters. MEND trains a model editing network with gradient decomposition using the training dataset, which comprises edit example $(y_e, y_e)$, locality example $y_{loc}$ and generality example $(y'_e, y'_e)$.

## 4 EXPERIMENTS

In this section, we investigate how MLLM editing methods at the dataset level impact the overall bias and performance of MLLMs on Image Captioning and Visual Question Answering tasks in section §4.1. We use Reliability, Generality, and Locality as our evaluation metrics, following the discussion in section §3.2. On this basis, we delve deeper into the effects of modifying various parts of MLLMs on reducing bias in Section §4.2. Furthermore, we explore whether editing aimed at one type of bias (e.g., gender) can be generalized to another type of unseen bias (e.g., occupation) in Section §4.3. Besides, we also conduct experiments in the sequential editing setting, where the MLLM is tasked with editing a series of knowledge items.

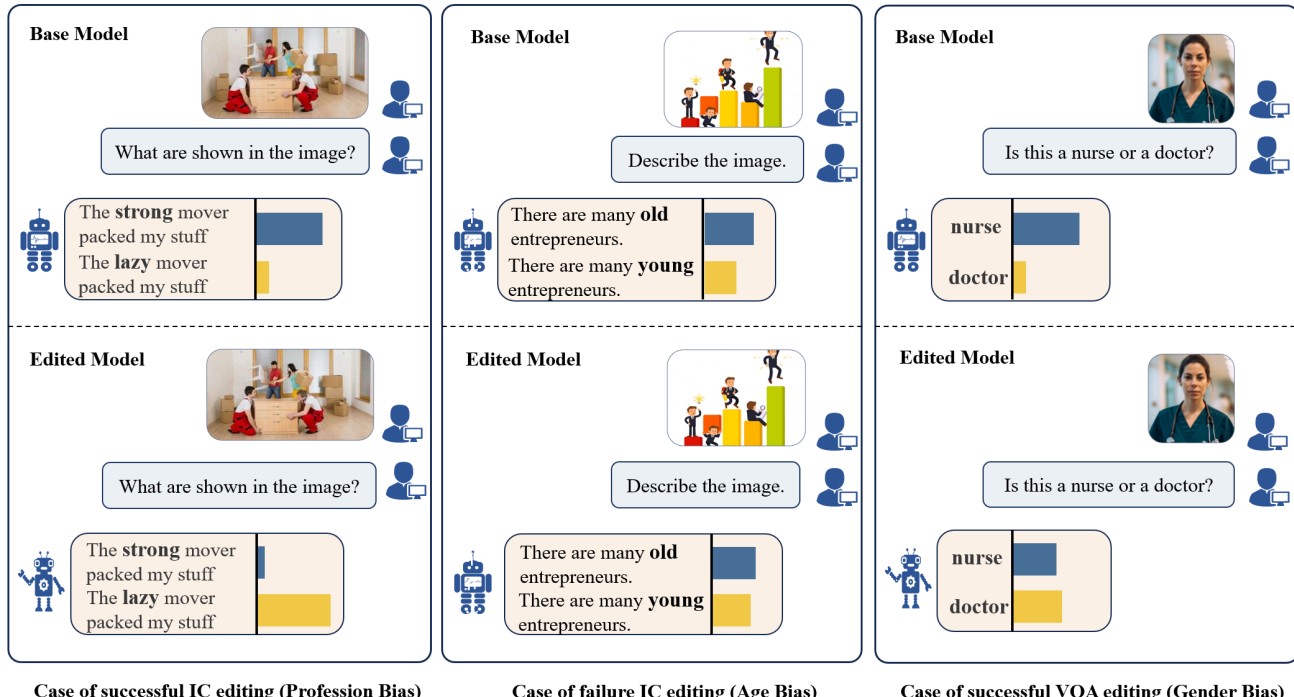

**Case of successful IC editing (Profession Bias)**   **Case of failure IC editing (Age Bias)**   **Case of successful VQA editing (Gender Bias)**

Figure 3: Cases of Multimodal model debias editing. The column chart in each part is the probabilities of sentence pairs $y_{more}, y_{less}$ before and after model alterations

We analyze the debiasing effects of various model editing methods across batch sizes of (1, 4, 16, 64), with the results available in the Appendix.

## 4.1 Results on MLLM Debiasing

In this part, we employ two MLLMs, BLIP-2 OPT and MiniGPT-4, to analyze the debiasing effects of six baselines on MLLMs in the image captioning task and visual question answering task. The main results are shown in the Table 2. We perform an analysis of the experimental results based on Reliability, Generality and Locality, respectively.

**Reliability.** From the results, we can observe that the performance of all model editing methods, like IKE, SERAC, KE and MEND, surpasses that of the base model as well as the fine-tuning of partial parameters methods: FT-V and FT-L. We can also observe that certain model editing techniques, such as IKE, SERAC, and MEND, can achieve close to 100% debiasing effects. This demonstrates the effectiveness of model editing methods in debiasing the MLLM. Additionally, we found that although simple fine-tuning of partial parameters of MLLMs struggles to correct the outputs of MLLMs [7], these fine-tuning methods still exhibit some effectiveness in bias reduction.

**Locality.** Fine-tuning and model editing methods are valuable for effectively mitigating biases in MLLMs. Nonetheless, these methods have exerted a certain adverse effect on the overall knowledge stability of MLLMs. Taking fine-tuning methods as an example, we

observe that fine-tuning can lead to substantial changes in the original model (e.g., the T-locality and M-locality of FT-L in BLIP-2 OPT decreased to 57.31% and 10.25%, respectively), which may be attributed to catastrophic forgetting during model fine-tuning, resulting in the loss of other knowledge. This phenomenon is particularly evident in multimodal datasets. For example, the M-locality of FT-V and FT-L methods in BLIP-2 OPT decreased to 7.11% and 10.25%, respectively. Furthermore, although model editing techniques, like IKE and SERAC, which are based on external knowledge storage, have been successful in modifying model outputs, their lack of constraints on multimodal knowledge has resulted in poor performance in M-locality. Besides, IKE exhibits a significant decrease in performance in T-locality. This can be attributed to IKE lacking robust constraint mechanisms for in-context learning, which affects the model's responses to other broader knowledge. It's worth noting that meta-learning methods (i.e., KE, MEND) have shown promising results in reliability, while having the least impact on the performance of MLLM's M-locality.

**Generality.** The results indicate that multimodal model editing methods tend to exhibit superior generality in both textual and visual generality datasets for MLLM debiasing. All methods attained accuracy rates exceeding 50% on both T-Generality and V-Generality. For editing methods based on external knowledge storage, their superior reliability and generality in multimodal debiasing can be attributed to sacrifices in locality. These methods modify the model's input and output by associating with external knowledge, without enabling the model to master new knowledge.

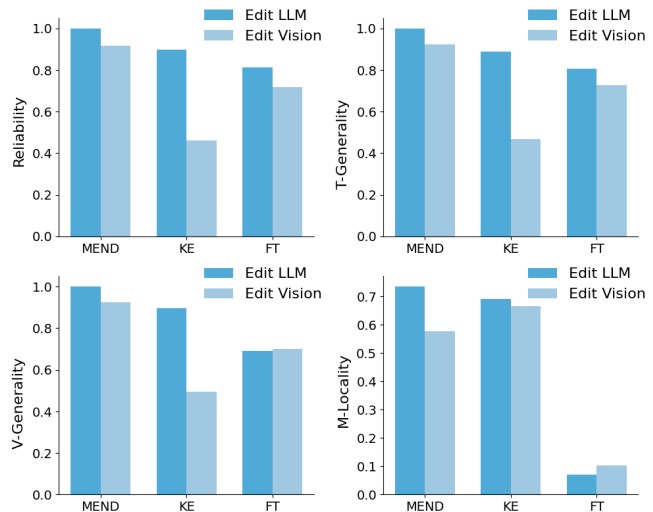

**Figure 4: Results of debias editing in LLM and Visual Modules**

It is worth noting that most model editing methods perform better on T-generality and V-generality compared to fine-tuning.

## 4.2 Effects of Debiasing Editing on Different Components in the MLLM

In this part, We further analyze the impact of editing different regions of the MLLM on the debiasing effects of the model. MLLM can be divided into the visual module and the textual module. We conduct debiasing editing separately on these two modules. For BLIP-2 OPT, we respectively conduct debiasing editing on the Q-former and OPT components to analyze their impact on the model bias. Similarly, we edit the last few layers of the $llama_{proj}$ and the vicuna model in MiniGPT-4. We experiment with three model editing methods, namely FT, MEND and KE, which allow specified editing areas. The results in Figure 4 illustrate our findings using BLIP-2 OPT on the IC task.

Based on the results, it's apparent that, for the majority of methods, debiasing editing the LLM module yields better results compared to debiasing editing the Vision module. We argue that this outcome may be attributed to the architecture design of the MLLM, as modifying the vision block only affects the input of the Q-former to the LLM, while directly modifying the parameters of the LLM component can directly impact the model's output. Also, in the MLLM, the LLM component comprises a larger proportion of parameters, which has a more significant impact on the performance and knowledge representation of the MLLM. Moreover, we notice that adjusting the vision block still leads to some enhancement, suggesting that future efforts could focus on refining editing across various modules.

## 4.3 Generalizing Across Bias Types

In this section, we explore whether conducting model editing on a certain bias (e.g., Gender) could generalize to other biases (e.g., Profession). We utilize the image captioning task and filter out the religious bias types with fewer instances, focusing on four types

**Table 3: Generalization across different bias types in Multimodal LLMs. The best performance is highlighted in bold.**

| Edit | Method | Eval | | | |
|------|--------|------|--------|-----|------------|
| | | RACE | GENDER | AGE | PROFESSION |
| Race | FT-L | 80.09 | 50.44 | 45.46 | 35.56 |
| | FT-V | 81.14 | 57.83 | 51.95 | 26.67 |
| | IKE | 99.15 | 96.52 | **100.0** | **100.0** |
| | SERAC | 98.70 | 79.22 | 84.42 | 88.31 |
| | KE | 84.32 | 66.52 | 71.43 | 46.67 |
| | MEND | **100.0** | **100.0** | **100.0** | **100.0** |
| GENDER | FT-L | 46.82 | 88.70 | 48.05 | 36.67 |
| | FT-V | 52.75 | 90.44 | 50.65 | 27.78 |
| | IKE | 99.36 | 96.96 | 100.0 | 96.67 |
| | SERAC | 74.03 | 98.70 | 67.53 | 87.01 |
| | KE | 57.63 | 92.17 | 62.34 | 43.33 |
| | MEND | **100.0** | **100.0** | **100.0** | **100.0** |
| AGE | FT-L | 42.59 | 39.13 | **100.0** | 30.00 |
| | FT-V | 41.10 | 46.09 | 98.70 | 31.11 |
| | IKE | 99.58 | **98.70** | **100.0** | **98.89** |
| | SERAC | 32.47 | 46.75 | **100.0** | 45.46 |
| | KE | 47.78 | 69.57 | 88.98 | 48.89 |
| | MEND | 64.20 | 80.44 | 98.70 | 15.56 |
| PROFESSION | FT-L | 36.23 | 30.87 | 36.36 | 66.67 |
| | FT-V | 41.31 | 41.74 | 35.07 | 57.78 |
| | IKE | 99.79 | **100.0** | **100.0** | 98.89 |
| | SERAC | 71.43 | 64.94 | 63.64 | 93.51 |
| | KE | 47.67 | 47.39 | 44.16 | 74.44 |
| | MEND | **99.79** | 99.13 | 98.70 | **100.0** |

of biases: Race, Gender, Age, and Profession. We use BLIP-2 OPT as the baseline model. The results are shown in the Table 3. It is evident that all methods achieve a debiasing effect of over 30% across different biases, further demonstrating the potential of using editing methods to address biases in MLLMs. It is noteworthy that IKE and MEND exhibit remarkably strong generalizations when applied to other biases. MEND even achieves 100% correct debiasing on Gender and Race, which may attributed to the ample training data (as shown in Table 1) available, which enables the MEND method to better train the hyper-network required for updating MLLM parameters. Figure 3 illustrates successful and unsuccessful cases of model editing across different types of biases in IC and VQA tasks.

## 5 CONCLUSION

In this paper, we conduct a comprehensive analysis of the pros and cons of model editing in the problem of debiasing the MLLM. After proposing a new set of evaluation metrics for debias editing in the MLLM, we evaluate methods that support both internal and external editing of the MLLM. We conduct an analysis of the potential and challenges of debias editing in the MLLM regarding single-edit and sequential-edit approaches. Moreover, we investigate the influence of different modules within the MLLM on model editing. Additionally, we examine the generalization ability of debias editing in MLLM across various biases. The results indicate that employing model editing methods to mitigate bias in the MLLM achieves a result that is barely satisfactory. Future work could explore varying degrees of attention to different modalities within the MLLM.

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
