# OpenReview forum: "Can We Debiase Multimodal Large Language Models via Model Editing?"
_acmmm.org/ACMMM/2024/Conference — MM2024 Poster_

### Official Review · Reviewer_eyjZ · 2024-05-22

**Rating:** 5
**Confidence:** 3

**Summary:**

This paper studies the topic of debiasing MLLMs by model editing. A benchmark dataset is built and several model editing methods are compared on the task.

**Strengths:**

1) Model editing is an emerging area of LLM research. This paper studies the feasibility of applying model editing to debiasing MLLMs. The idea is novel to me.
2) Comprehensive evaluation metrics and a benchmark dataset are built which facilitate following studies on this topic.

**Limitations:**

1) In addition to image captioning and VQA, more downstream tasks are expected to be included in the evaluation.
2) The remaining challenges of applying model editing to debiasing MLLMs should be discussed.

**Suitability:**

3

---

### Official Review · Reviewer_aZsU · 2024-05-24

**Rating:** 3
**Confidence:** 2

**Summary:**

The paper discusses the problem of biases in multimodal large language models (MLLM) and proposes the use of model editing techniques to address this issue.

**Strengths:**

1.The paper introduces a comprehensive set of evaluation metrics, including Reliability, Generality, and Locality, designed to assess the debiasing capabilities of multimodal large language models (MLLM).
2.MEND achieves 100 for many metrics in experimental results.

**Limitations:**

1.This paper is similar to "Can We Edit Multimodal Large Language Models?", especially the figures.
2.The experiment results are different from "Can We Edit Multimodal Large Language Models?" on the same baselines and metrics.
3.the sentence "This picture depicts" seems a little strange in Figure1

**Suitability:**

3

---

### Official Review · Reviewer_H4dG · 2024-05-26

**Rating:** 4
**Confidence:** 3

**Summary:**

This paper explores the application of model editing techniques to mitigate biases in MLLMs. The authors introduce a benchmark for evaluating debiasing methods in MLLMs, focusing on reliability, generality, and locality, and conduct extensive experiments using two popular MLLM models, BLIP-2 and MiniGPT-4. The findings reveal the effectiveness of debiasing editing methods, particularly when applied to the textual module of MLLMs, and highlight the trade-offs between reducing biases and maintaining other model capabilities.

**Strengths:**

- This paper proposes a debiasing benchmark that derived from existing datasets to evaluate the debiasing method quantitatively.
- The experiments show the effectiveness of existing debiasing method in editing MLLMs.

**Limitations:**

- For the reason that utilize model editing to alleviate the biases instead of other three paradigms (i.e., line 101 to line 114), do you have a quantitative example to directly show the difference and improvements?
- Unclear description in Sec. 3.1 and Sec 3.2. What does the $y$ in the formula (1) and (2) refer to with an VQA sample? What do $y_more$ and $y_less$ look like?  Could you provide an example in comparison with standard unbiased scene? Additionally, how do we define that a MLLM exhibits bias towards an image-text pair? Does the bias mean that in any situation the MLLM tend to consider such image-text pair?
- As introduced at line 516, where is the Appendix?
- Which version of the Stable Diffusion model that you used in this study? SD1, SD2 or SDXL?
- The MLLMs used in the experiment for model editing are out of date. The current typical models should be considered, such as LLaVA series, CogVLM series, DeepSeek-VL, Qwen-VL-Chat, VILA.
- No case studies to show the typical biased examples that could be modified effectively.


typos:
  - at line 435, presents -> represents

lacking of references for MLLMs:
  - Bai, Jinze, et al. "Qwen-vl: A frontier large vision-language model with versatile abilities." arXiv preprint arXiv:2308.12966 (2023).
  - Lu, Haoyu, et al. "DeepSeek-VL: towards real-world vision-language understanding." arXiv preprint arXiv:2403.05525 (2024).
  - Wang, Weihan, et al. "Cogvlm: Visual expert for pretrained language models." arXiv preprint arXiv:2311.03079 (2023).
  - Qi, Ji, et al. "CogCoM: Train Large Vision-Language Models Diving into Details through Chain of Manipulations." arXiv preprint arXiv:2402.04236 (2024).
  - Lin, Ji, et al. "Vila: On pre-training for visual language models." arXiv preprint arXiv:2312.07533 (2023).

**Suitability:**

3

---

### Official Review · Reviewer_4NPZ · 2024-05-27

**Rating:** 4
**Confidence:** 4

**Summary:**

This paper proposes to evaluate the effectiveness of model editing methods for debiasing multimodal large language models. To this end, three evaluation metrics, including Reliability, Generality, and Locality, and a debiasing benchmark, are introduced.

**Strengths:**

1. It is very meaningful to explore the impact of model editing on debiasing multimodal large language models.
2. The introduced evaluation metrics and benchmarks can promote the elimination of bias in multimodal large language models.
3. Evaluated model editing methods are comprehensive.

**Limitations:**

About M-Locality, is a simple OK-VQA enough to test that the overall ability of MLLM is not affected after editing?

**Suitability:**

3

---

### Meta-Review · Area_Chair_MGrc · 2024-07-08

**Recommendation:** Accept (Poster)
**Confidence:** 4

**Metareview:**

The paper presents some advances in mitigating biases in multimodal large language models (MLLMs) through model editing. It introduces a comprehensive benchmark for evaluating debiasing methods, focusing on reliability, generality, and locality. Reviewers praised the novelty and importance of the study, highlighting the clear motivation, comprehensive evaluation metrics, and the effectiveness demonstrated in experiments. The paper’s introduction of model editing as a debiasing approach is particularly noteworthy, offering a novel and environmentally friendly fine-tuning strategy. While some minor areas for improvement were identified, such as expanding the range of downstream tasks and clarifying certain methodological details, these do not detract from the overall value and contribution of the work. The reviewers are confident in the paper's suitability for presentation and believe it will significantly contribute to the field of multimedia/multimodal processing. With minor revisions to address the detailed feedback, this paper is well-positioned for acceptance.